# Notch2-mediated plasticity between marginal zone and follicular B cells

Markus Lechner[1], Thomas Engleitner[2,3,4], Tea Babushku [1], Marc Schmidt-Supprian[2,4,5], Roland Rad [2,3,4], Lothar J. Strobl [1,6] & Ursula Zimber-Strobl [1,6✉]

Follicular B (FoB) and marginal zone B (MZB) cells are functionally and spatially distinct mature B cell populations in the spleen, originating from a Notch2-dependent fate decision after splenic influx of immature transitional B cells. In the B cell follicle, a Notch2-signal is provided by DLL-1-expressing fibroblasts. However, it is unclear whether FoB cells, which are in close contact with these DLL-1 expressing fibroblasts, can also differentiate to MZB cells if they receive a Notch2-signal. Here, we show induced Notch2IC-expression in FoB cells reprograms mature FoB cells into bona fide MZB cells as is evident from the surface phenotype, localization, immunological function and transcriptome of these cells. Furthermore, the lineage conversion from FoB to MZB cells occurs in immunocompetent wildtype mice. These findings demonstrate plasticity between mature FoB and MZB cells that can be driven by a singular signaling event, the activation of Notch2.

[1] Research Unit Gene Vectors, Helmholtz Zentrum München GmbH, German Research Center for Environmental Health, München, Germany. [2] TranslaTUM, Center for Translational Cancer Research, Technical University of Munich, München, Germany. [3] Institute of Molecular Oncology and Functional Genomics, TUM School of Medicine, Technical University of Munich, Munich, Germany. [4] German Cancer Consortium (DKTK), Heidelberg, Germany. [5] Institute of Experimental Hematology, TUM School of Medicine, Technical University of Munich, Munich, Germany. [6] These authors contributed equally: Lothar J. Strobl, Ursula Zimber-Strobl. ✉email: strobl@helmholtz-muenchen.de

FoB and MZB cells are two distinct mature B lymphocyte populations that differ in their immunological function and anatomical localization. FoB cells recirculate between lymphoid organs and give rise to germinal center B cells that undergo class switching and somatic hypermutation to produce high-affinity antibodies and memory B cells in T cell-dependent immune reactions. In sharp contrast, murine MZB cells are sessile in the name-giving marginal zone (MZ) of the spleen and their migration is limited to the shuttling between the MZ and the splenic B cell follicle[1,2]. MZB cells are responsible for the first line defense against blood-borne pathogens and rapidly produce low-affinity IgM antibodies against encapsulated bacterial and poly-saccharide antigens in T cell-independent immune reactions[3]. They are therefore commonly referred to as innate B cells[4].

The current model suggests that immature B cells leave the bone marrow and migrate to the spleen as transitional B cells where they undergo a cell fate decision to become either FoB or MZB cells[5–10]. However, it has also been suggested that MZB cells originate from FoB cells. Thus in rats, FoB cells have been shown to act as immediate precursors for MZB cells[11,12]. But the differentiation of MZB cells in rats and mice may not be completely comparable, because unlike in mice MZB cells are the major mature B cell population in rats[13]. It was also shown that transplantation of FoB cells into immunodeficient mice leads to the generation of MZB cells[9,14,15]. Yet, this phenotype could not be observed in immunocompetent hosts[9]. It has been suggested that the differentiation of FoB to MZB cells is due to the lymphopenic mice in which the transferred B cells are under immediate antigenic pressure, resulting in their activation accompanied by a MZB-like phenotype and PC differentiation[16]. So, it is still an open question, whether FoB cells act as precursors for MZB cells in immunocompetent mice.

Moreover, it is still elusive how different signaling pathways interact to commit B cells toward the MZB cell fate. The phenotypes of different transgenic mouse models led to the assumption that a weak BCR signal promotes differentiation of MZB cells while strong BCR signaling directs transitional B cells toward the FoB cell compartment[5,10]. This assumption, however, is controversially discussed and some mouse mutants do not fit in this scenario[17]. According to the current knowledge BCR-, BAFF-R-, canonical-NF-κB- and Notch2-signaling work together in inducing MZB cell development[5,17]. Conditional inactivation of Notch2 in B lymphocytes or its ligand Dll-1 in non-hematopoietic cells results in loss of MZB cells, indicating that Notch2 signaling is essential for the differentiation of MZB cells.[18,19]. Interaction of the Notch2 receptor with its ligand DLL-1 on a neighboring cell results in two proteolytic cleavages of Notch2 followed by translocation of the intracellular part of the receptor (Notch2IC) into the nucleus where it interacts with the DNA-binding protein RBPJ to regulate gene expression[20]. We have shown recently that constitutive active Notch2-signaling in B cells strongly shifts transitional type 1 (T1) B cells toward the MZB cell lineage, even in the absence of CD19[21]. The timing and location of Notch2 activation during MZB cell development harbors some ambiguity. It has been suggested for a long time that activation of Notch2 takes place in the splenic red pulp[22]. This assumption was challenged by recent work demonstrating that DLL-1 expression on follicular fibroblasts is crucial for MZB cell development[23], suggesting that Notch2-signaling is induced within the follicular B cell zone. Since the B cell compartment is a dynamic micro-environment[2], it can be assumed that FoB cells can get into contact with follicular fibroblasts and thereby receive DLL-1 mediated activation of Notch2. Considering the strong differentiation inducing potency of Notch2-signaling in T1 cells toward MZB cells[21], we wondered whether FoB cells start to differentiate to MZB cells if they receive an above-threshold Notch2-signal.

To answer this question, we induced Notch2IC-expression exclusively in mature FoB cells and followed the fate of these cells over time. Here, we show that Notch2-signaling is sufficient to induce a complete trans-differentiation of mature FoB cells into MZB cells in vivo. By transplanting mature FoB cells in immunocompetent mice, we were able to verify a remarkable plasticity between FoB and MZB cells also in a non-transgenic setting.

## Results

**Induction of Notch2-signaling in mature FoB cells**. To analyze whether Notch2-signaling can induce a phenotypic shift from FoB toward MZB cells, the Notch2IC[flSTOP] strain[21] was combined with homozygous CD19-creER[T2hom] mice[24], in which Cre activity can be induced specifically in B cells by tamoxifen. Untreated Notch2IC[flSTOP]/CD19-creER[T2hom] mice (termed N2IC/creER[T2hom] hereafter) lack MZB cells due to their CD19-deficiency caused by the CreER[T2hom] insertion on both Cd19 alleles (Fig. 1a)[25]. This enabled us to trace the appearance of newly generated MZB cells in the absence of pre-existing MZB cells. To exclude any effect of transitional B cells, we inhibited the influx of immature B cells from the bone marrow (BM) by four injections of an inhibitory antibody against the interleukin 7 receptor (IL7-R)[26] (Fig. 1b, Supplementary Fig. 1). Cre-mediated Notch2IC/hCD2-expression was induced by application of a single dose of tamoxifen. The complete experimental setup is shown in Fig. 1b. As controls, we used CAR[flstop]/CD19-creER[T2hom] (termed CAR/creER[T2hom] hereafter) reporter mice[27]. These mice express a truncated version of the human coxsackie/adenovirus receptor (CAR) upon Cre-mediated recombination (R26/CAG-CARΔ1[StopF]). The CAR receptor can be stained at the cell surface by FACS. As further control heterozygous Notch2-IC[flSTOP]/CD19-creER[T2het] mice, carrying one intact Cd19 allele (termed N2IC/creER[T2het] hereafter) were included in the analysis.

**Notch2IC-expression induced phenotypic conversion of FoB to MZB cells**. Administration of tamoxifen resulted in elimination of the loxP-flanked STOP cassette and subsequent expression of Notch2IC together with the surface reporter human CD2 (hCD2) in around 1% of FoB cells after 3 days. The percentage of Notch2IC-expressing hCD2[+] B cells peaked two weeks after tamoxifen application reaching around 2–7% of total B cells (Fig. 1c). Hence, the cell pool of hCD2[+] cells resembled the physiological size of the MZB cell compartment. To analyze whether Notch2IC-expressing FoB cells shift their phenotype to MZB cells, the CD23/CD21 surface expression of hCD2[+] splenic B cells was followed over time. hCD2[+] cells revealed a slow but almost complete shift from CD23[+]CD21[high] on day 3 to CD23[low]CD21[high] on day 14 (Figs. 1d, e). In parallel, hCD2[+] cells increased their size (FSC), upregulated IgM and CD1d and downregulated IgD (Fig. 1f). Thus, throughout our observation period the phenotype of Notch2IC-expressing cells gradually acquired a MZB cell surface phenotype (CD23[low]CD21[high]-IgM[high]IgD[low]CD1d[high]). Repeated doses of tamoxifen on five consecutive days significantly increased the percentage of hCD2[+] B cells (Supplementary Fig. 2a) and thus the percentage of CD23[low]CD21[high] B cells, but had no accelerating effect on the kinetic of the phenotypic change (Supplementary Fig. 2b). The percentage of hCD2[+] B cells peaked at week 2 at 25% and declined continuously reaching 1–4% 12 weeks after tamoxifen induction (Supplementary Fig. 2a). Induction of Notch2IC-expression in Notch2IC[flSTOP]/CD19creER[T2het] mice, which still possess one intact Cd19 allele, resulted in a phenotypic shift from FoB to MZB cells in comparable kinetic and frequency (Supplementary Fig. 2c and d) as in Notch2IC[flSTOP]/CD19creER[T2hom] mice. To determine whether the trans-differentiation of FoB to

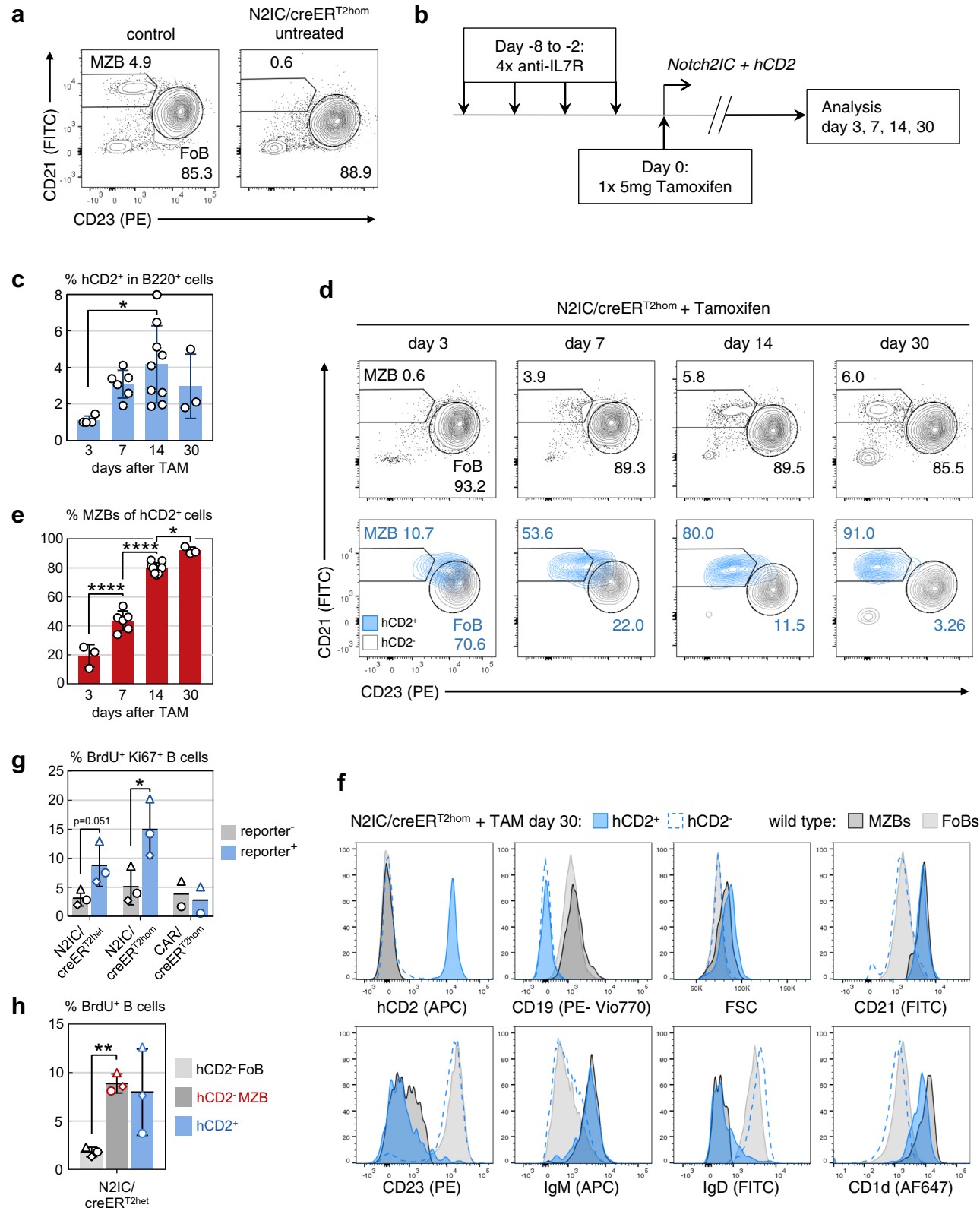

MZB cells is correlated with proliferation, tamoxifen-induced mice were fed with BrdU for 7 days. The percentage of BrdU+ Ki67+reporter+ splenic B cells in N2IC/creER$^{T2hom}$ and N2IC/creER$^{T2het}$ was higher in comparison to CAR/creER$^{T2hom}$ and reporter− splenic B cells (Fig. 1g). But even among the Notch2IC-expressing B cells, only a sub-fraction of B cells (~9% and 15%, respectively) incorporated BrdU. To analyse whether the proliferation of Notch2IC-expressing B cells is similar to that of MZB cells, we compared the percentages of BrdU+ B cells between reporter− MZB and FoB cells from N2IC/creER$^{T2het}$. In accordance with the previous observations[9,28], the percentage of BrdU+ cells was higher in MZB cells than FoB cells (Fig. 1h). Notably, in N2IC/creER$^{T2het}$ the percentages of BrdU+ in hCD2+ cells and hCD2− MZB cells were comparable, suggesting that the

**Fig. 1 Notch2IC drives conversion from a FoB toward a MZB surface phenotype. a** Representative FACS analysis of B220+ splenocytes from untreated N2IC/creER[T2hom] and control mice. Numbers represent the percentages of CD23[low]CD21[high] MZB and CD23+CD21+ FoB cells. $n > 6$. **b** Treatment scheme of N2IC/creER[T2hom] mice to induce Notch2IC/hCD2-expression in FoB cells. **c** Percentages of hCD2+ splenic B lymphocytes at the indicated time points after tamoxifen treatment. $*p = 0.0223$, d3 $n = 4$, d7 $n = 6$, d14 $n = 9$, d30 $n = 3$. **d** Representative FACS analysis for CD23[low]CD21[high] MZB cells at the indicated time points after tamoxifen administration. The plots are gated on B220+ lymphocytes (upper row) and in addition to hCD2− (gray) and hCD2+ (blue) B cells (lower row). Indicated percentages refer to the gated populations within all B cells (black) and hCD2+ cells (blue). Plots are representative for repeated biological independent experiments per time point with d3 $n = 4$, d7 $n = 6$, d14 $n = 9$, d30 $n = 3$ animals. **e** The graph compiles the percentages of CD23[low]CD21[high] MZB cells within hCD2+ cells at the indicated time points. $****p = 3.0E−05$ (d3 vs d7), $****p = 1.6E−09$ (d7 vs d14), $*p = 0.0144$. d3 $n = 3$, d7 $n = 5$, d14 $n = 7$, d30 $n = 3$. **f** Representative FACS analysis for the indicated MZB markers within hCD2+ and hCD2− splenocytes from N2IC/creER[T2hom] mice 30 days after tamoxifen treatment. MZB and FoB cells from control animals are shown in grayscales. $n = 3$. **g, h** Mice with the indicated genotypes were treated as shown in (**b**) and fed with BrdU via drinking water from day 1 to 7. FACS analysis for BrdU+ B cells was performed on day 7 after tamoxifen treatment ($n = 3$ for N2IC mice, $n = 2$ for CAR mice). **g** Percentages of BrdU/Ki67 double positive cells within reporter+ (hCD2+ or CAR+: blue) and reporter− B220+ splenocytes (gray) from mice with the indicated genotypes. All BrdU+ cells were also positive for Ki67. $*p = 0.0128$. **h** Percentages of BrdU+ cells within splenic hCD2− MZB (red) and hCD2− FoB cell (gray) in comparison to hCD2+ cells (blue) from N2IC/creER[T2het] mice. $**p = 0.0034$. **g, h** Same symbol shapes represent data from the same mouse within a genotype. For all bar graphs means, SDs and the values for each single mouse are indicated. **c, e** one-way ANOVA, Tukey's multiple comparisons test; **g, h** paired $t$-tests, two tailed.

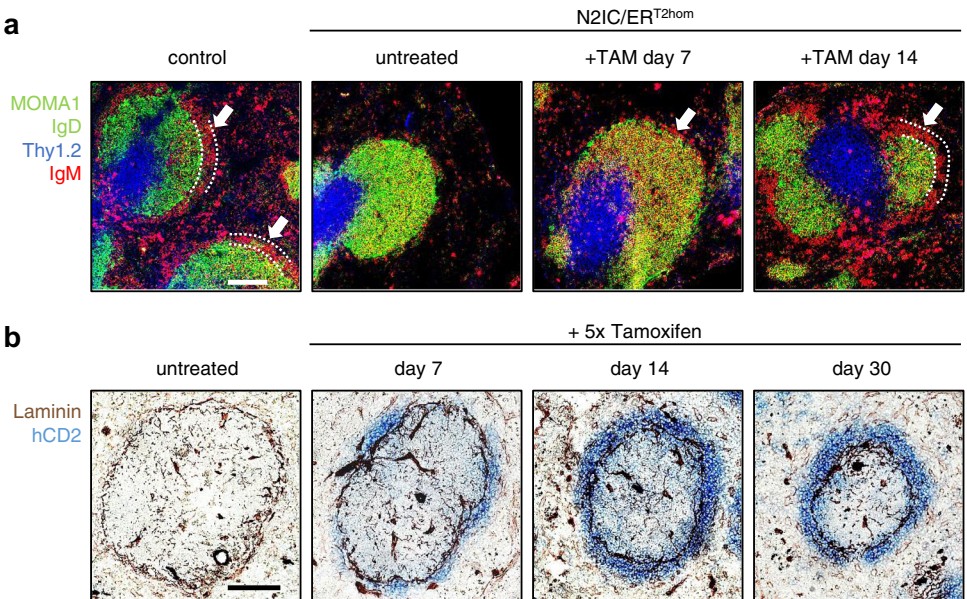

**Fig. 2 Notch2IC-expressing cells progressively repopulate the marginal zone. a** Immunofluorescence analysis of splenic sections from control and N2IC/creER[T2hom] animals, with or without tamoxifen (TAM) treatment for the indicated time points. Sections were stained for FoB cells (IgD+, green), metallophilic macrophages (MOMA-1+, green), T cells (Thy1.2+, blue), and MZB cells (IgM++, red). MZB cells are highlighted by white arrows, the marginal zone is additionally delineated with dotted white lines. **b** Chromogenic immunohistochemical analysis for hCD2 expression after tamoxifen treatments. N2IC/creER[T2hom] mice were treated with 5 mg tamoxifen on 5 consecutive days. Indicated time points represent days after initial treatment. Notch2IC-expressing cells were stained for hCD2 (blue). The basement membranes of endothelial cells lining the MZ sinus and other blood vessels in the white and red pulp were stained by an anti-Laminin antibody (brown). The scale bars represent 100 μm. Images are representative for splenic sections of 2 mice per group.

proliferation of Notch2IC-expressing B cells resembles the proliferation of MZB cells. These data suggest that constitutive active Notch2-signaling in FoB cells induces a phenotypic conversion from FoB toward MZB cells, at least with regard to the analyzed surface markers characteristic for MZB cells. Furthermore, our data show that Notch2IC-expression overcomes the CD19-dependency of MZB cell development.

**The newly generated MZB cells localized to the marginal zone.** MZB cells reside in a well-defined splenic microenvironment, the so-called marginal zone (MZ). This site is exclusive for MZB cells, as they are equipped with special traits to cope with the shear forces in the MZ, where they are exposed to blood-borne antigens and poised to rapidly respond to pathogens[10]. The blood-perfused MZ surrounds the compact lymphocyte-rich follicle,

which divides into a T cell zone and the B cell zone, harboring FoB cells. The follicle is demarcated by a ring-like laminin+ MZ sinus, bordered with MOMA1+ metallophilic macrophages[29]. As expected, IgM[high]IgD[low] MZB cells surrounding the marginal sinus and the ring of MOMA-1+ macrophages were detected in histological sections of control mice, but were missing in untreated N2IC/creER[T2hom] mice (Fig. 2a). Notably, 1 week after tamoxifen treatment MZB cells started to reappear at their defined anatomic location (highlighted with white arrows) and a complete ring structure of MZB cells was detected 14 days after induction of Notch2IC-expression. The progressively reappearing MZB cells expressed hCD2 and from day 14 after tamoxifen treatment onwards, most hCD2+ cells were localized in the MZ (Fig. 2b).

To sum up, Notch2IC-expression in FoB cells not only leads to a phenotypic shift of FoB to MZB surface markers, but also guides

the trans-differentiating MZB cells to their proper splenic localization in the MZ.

**Reconstituted MZB cells are hyperresponsive to LPS**. We then tested whether the newly trans-differentiated MZB cells functionally resemble their control counterparts. MZB cells are often termed "innate-like" lymphocytes because they come with a set of special traits, enabling them to rapidly respond to blood-borne pathogens[3]. MZB cells respond much faster than FoB cells to TLR stimulation in vitro. Accordingly, after short term stimulation with LPS, MZB cells show an enhanced proliferation and PC-differentiation in comparison to FoB cells[3,30–33]. To challenge the functionality of reconstituted MZB cells, we compared the proliferation and PC differentiation of untreated and tamoxifen-treated N2IC/creER[T2hom] mice after LPS treatment in vitro for 48 h. Expectably, unlike CD19-proficient control B cells, purified B cells from untreated N2IC/creER[T2hom] mice did not proliferate or differentiate to plasmablasts upon LPS stimulation in vitro due to the lack of MZB cells (Fig. 3a). In contrast, 14 days after tamoxifen induction, B cells of N2IC/creER[T2hom] mice displayed a robust plasmablast differentiation and proliferation (Fig. 3b). This response was exclusively mediated by hCD2[+] cells, suggesting an equivalent behavior of control and reconstituted MZB cells upon LPS-stimulation in vitro.

In contrast to FoB cells, MZB cells rapidly produce plasmablasts upon immunization with TI-antigens within 3 days[3]. To substantiate the functionality of reconstituted MZB cells in vivo, mice were immunized with the TI-1 antigen NP-LPS and analyzed for plasmablast differentiation after three days. In accord with our in vitro assays, untreated N2IC/creER[T2hom] mice were unable to generate plasmablasts upon NP-LPS immunization, whereas after tamoxifen-treatment, N2IC/creER[T2hom] mice responded to the NP-LPS challenge with production of plasmablasts in a similar manner as CD19-proficient control animals (Fig. 3c, d, Supplementary Fig. 3a–c). The high percentage of hCD2[+] B cells that differentiated to plasmablasts confirmed the high sensitivity of Notch2IC-expressing B cells to the NP-LPS challenge (Fig. 3e, Supplementary Fig. 3b). In addition, almost all CD138[+]TACI[+] and B220[low]IRF4[+] cells expressed hCD2 (Fig. 3e, Supplementary Fig. 3b), suggesting that the trans-differentiated MZB cells are exclusively responsible for the in vivo response to NP-LPS immunization. Both control plasmablasts and those derived from transdifferentiated MZB cells were IgM[+] and did not switch to IgG1 (Fig. 3f, g).

Collectively, these data show that the reconstituted Notch2IC-expressing MZB cells rapidly differentiate to plasmablasts upon LPS-challenge in vitro and in vivo and thus functionally resemble wild type MZB cells.

**Notch2 induces a transcriptomic shift from FoB to MZB signature**. To analyze the identity conversion of FoB to MZB cells on transcriptomic level we performed RNA sequencing (RNA-Seq). For the RNA-Seq analyses tamoxifen-induced hCD2[+] and hCD2[-] non-responding splenic B220[+] cells from N2IC/creER[T2hom] mice were sorted 3, 5, 7, and 14 days after tamoxifen-treatment. FoB and MZB cells from wild type mice served as controls. Unbiased iterative multidimensional scaling analysis (MDS) showed good clustering of sample groups and illustrated overall transcriptomic distances of clustered populations (Fig. 4a). The MDS analysis visualized the transcriptomic distance of wild type FoB and MZB populations. The distance and direction of the transcriptomic shift from hCD2[-] to hCD2[+] cells at day 14 cells was in parallel to the FoB to MZB axis. Neither wild type FoB and hCD2[-] cells nor wild type and trans-differentiated MZB cells clustered together. This

basic offset is most likely due to the CD19-deficiency of N2IC/creER[T2hom] mice, in comparison to wild type mice.

To verify the FoB-to-MZB cell conversion more specifically, MZB signature gene sets were defined by identifying the top 1000 differentially expressed genes (by p-value) of wild type MZB versus FoB cells from our RNA-Seq data and overlapped with the top 1000 differentially expressed MZB versus FoB cell-enriched genes provided by the ImmGen database[34] (Supplementary Fig. 4a and Supplementary Table 1). The thus created gene sets were termed "FoB genes" (comprised of 210 common genes, significantly downregulated in MZB versus FoB cells) and "MZB genes" (176 genes significantly upregulated in MZB versus FoB cells). A heatmap for all signature genes was created through normalizing their expression values across all samples of the Notch2IC-induction kinetic and the controls. The heatmap illustrates the stepwise loss of FoB and simultaneous gain of MZB cell-specific gene expression over time (Fig. 4b). While non-responding cells (hCD2[-]) kept a FoB cell signature, hCD2[+] cells adopted the MZB cell signature over time and closely resembled the wild type MZB cell transcriptome 14 days after induction of Notch2IC-expression. Additional gene set enrichment analysis (GSEA) proved the strong enrichment of signature genes at the very top (MZB genes) and bottom ends (FoB genes) of the ranked list of all differentially expressed genes between hCD2[+] and non-responding hCD2[-] cells (Fig. 4c). The gene set enrichment of the MZB signature was significant at each time point after induction of Notch2IC–expression and the enrichment score increased over time (Supplementary Fig. 4b). Visualizing all top differentially regulated genes by p-value and fold changes between MZB/FoB as well as hCD2[+] (day 14 after tamoxifen)/hCD2[-] sample pairs in volcano plots (Supplementary Fig. 5a and b) showed that many genes among the top-hits were found in both wild type and N2IC/creER[T2hom] sample pairs. Subsequent plotting of the fold changes of wild type MZB versus FoB cells against hCD2[-] versus hCD2[+] cells at day 14 identified a high correlation of top-up- and top-downregulated common genes (Supplementary Fig. 5c and d). Collectively, the RNA-Seq analysis revealed that constitutive Notch2-signaling in FoB cells initiated a substantial change of the transcriptome over a time course of 2 weeks, until a complete MZB cell gene expression signature was adopted.

**Notch2 regulates genes involved in transcription and homing**. To learn more about the regulation of Notch2IC-induced genes, we analyzed which transcriptional regulators are induced or repressed by Notch2IC and thus might contribute to the differential gene expression profile. Among the top upregulated genes were the known Notch-target genes *Hes1* and *Hes5*, which act as transcriptional repressors[35] as well as *Bcl7a*, a component of the chromatin-remodeling SWI/SNF complex[36]. Three transcription factors (*Irf8*, *Foxo1*, and *Klf2*), whose genetic inactivation leads to the expansion of the MZB cell compartment[37–41], were significantly downregulated.

Since *Klf2* was by far the most significantly and strongly downregulated gene, we analyzed the impact of its downregulation on the Notch2IC-gene expression profile in more detail. GSEA analysis using a KLF2 signature gene set defined by microarray data of *Klf2*-deficient FoB cells published by Hart and colleagues[39] (Supplementary Table 2) revealed strong and significant enrichment of up- and downregulated KLF2 target genes (Fig. 5b). These data suggest that the downregulation of *Klf2* significantly contributes to the differential gene expression profile between hCD2[+] and hCD2[-] B cells. In addition, several genes controlling migration and adhesion were found among the top Notch2IC-regulated genes (Fig. 5c). Sphingosine-1 phosphate receptor genes *S1pr1* and *S1pr3* were strongly upregulated, while

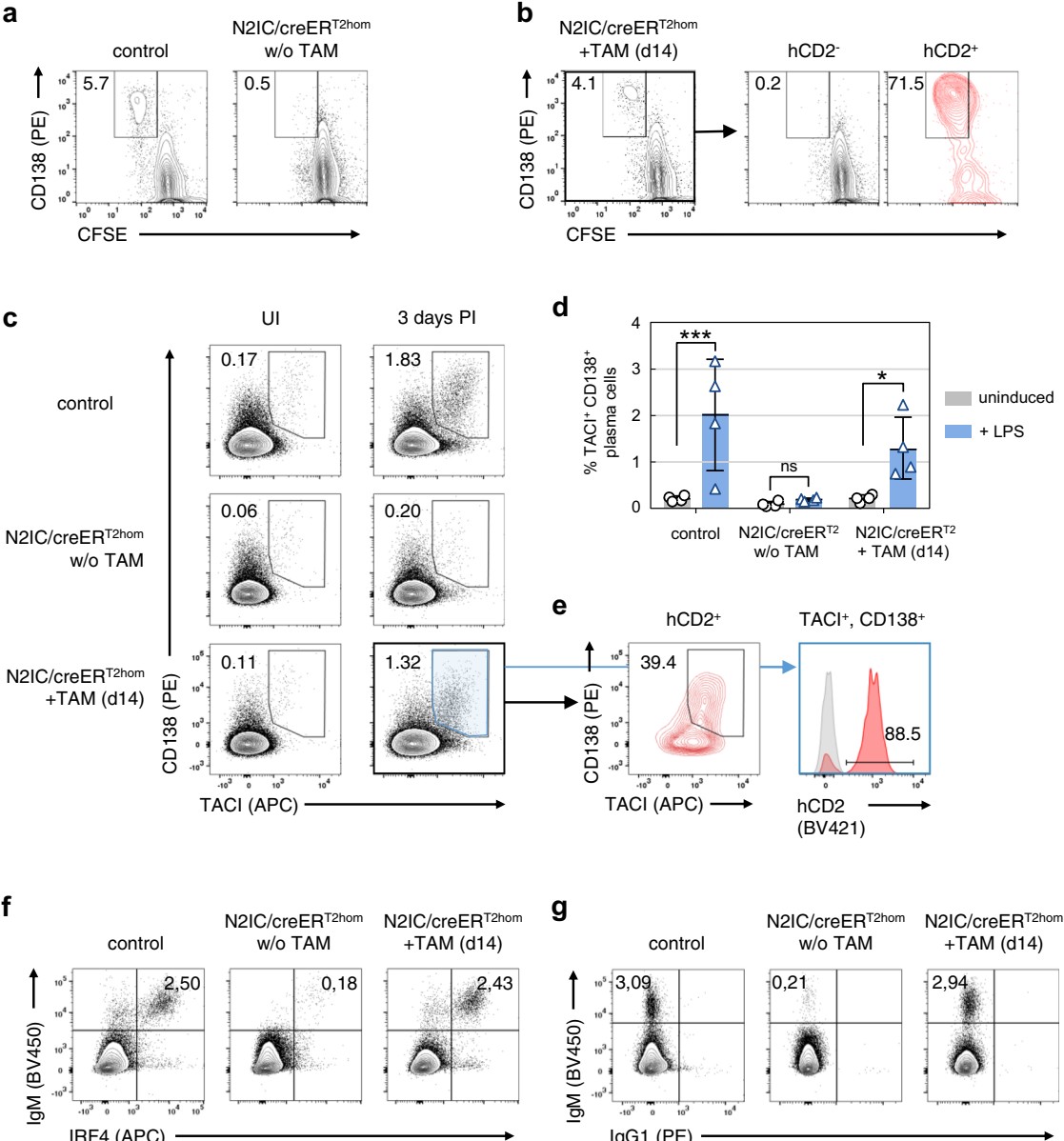

**Fig. 3 Reconstituted MZB cells functionally resemble wildtype MZB cells. a , b** Stimulation of purified B cells, with the indicated genotypes and treatment, with LPS for 48 h in vitro (w/o TAM: without tamoxifen treatment; +TAM (14d): day 14 after tamoxifen treatment). **a** Plots are pre-gated on living singlets, plasmablasts are gated as CD138[+]CFSE[low] cells. **b** Cells were gated on living singlets (left) and consecutively separated in hCD2[−] (black) and hCD2[+] (red) cells. FACS plots in **a**, **b** are representative for two independent experiments. **c–e** Plasmablast differentiation in vivo after immunization with NP-LPS: **c** flow cytometric analyses show the percentages of plasmablasts (CD138[+]TACI[+]) in the spleen of unimmunised (UI) and immunized control and N2IC/creER[T2hom] mice (3 days post immunization (PI) with NP-LPS). Cells were pregated on a large lymphocyte gate. **d** Summary and quantification of plasmablast percentages 3 days after NP-LPS immunization. Bar graph shows mean values and SD. Data points represent single mice from two independent experiments. ***$p = 8.0E-04$; *$p = 0.040$ (two-way ANOVA, Sidak´s multiple comparisons test). **e** Additional analysis of splenocytes from TAM-treated N2IC/creER[T2hom] mice after immunization with NP-LPS: Indicated are the percentage of plasmablasts (CD138[+]TACI[+]) within hCD2[+] cells and the percentage of hCD2[+] cells in all TACI[+]CD138[+] plasmablasts (red plot and histogram, respectively). Gray histogram: hCD2-staining of plasmablasts from immunized control animals (negative control). **f, g** Intracellular flow cytometric analysis of splenocytes from the immunized mice shown in (**c**). **f** Plasmablasts were identified as IRF4[high]IgM[high]. **g** Separation of plasmablasts in IgM[+] and IgG1[+] cells. FACS plots in **c–e** are representative for two individual experiments with $n = 4$ mice per group.

the family member *S1pr4* was downregulated. It is well accepted that the migration of MZB cells toward the marginal zone is mainly controlled by the S1PR1 and S1PR3[2,42], while S1PR4 has been suggested to modulate the S1PR1 induced signaling in B cells[43]. Besides, *Sema7a* and *Plxnd1* were strongly upregulated by Notch2-signaling. Plexin/Semaphorin interactions originally described to have a role in axon guidance during neuronal development are also involved in the activation or differentiation

of immune cells as well as their trafficking[44]. Semaphorin7a has not been described in the context of MZB cell migration, but inactivation of *Plexin-D1* results in reduced numbers of MZB cells[45]. Vice versa, *Cxcr5*, and *Ccr7*, encoding for two chemokine receptors that guide B cells into the B cell follicle and to the B/T cell border, respectively[46] were significantly downregulated. The known KLF2-target genes *Itgb7* and *Sell* might be downregulated by the repression of KLF2[37−39]. These data indicate that

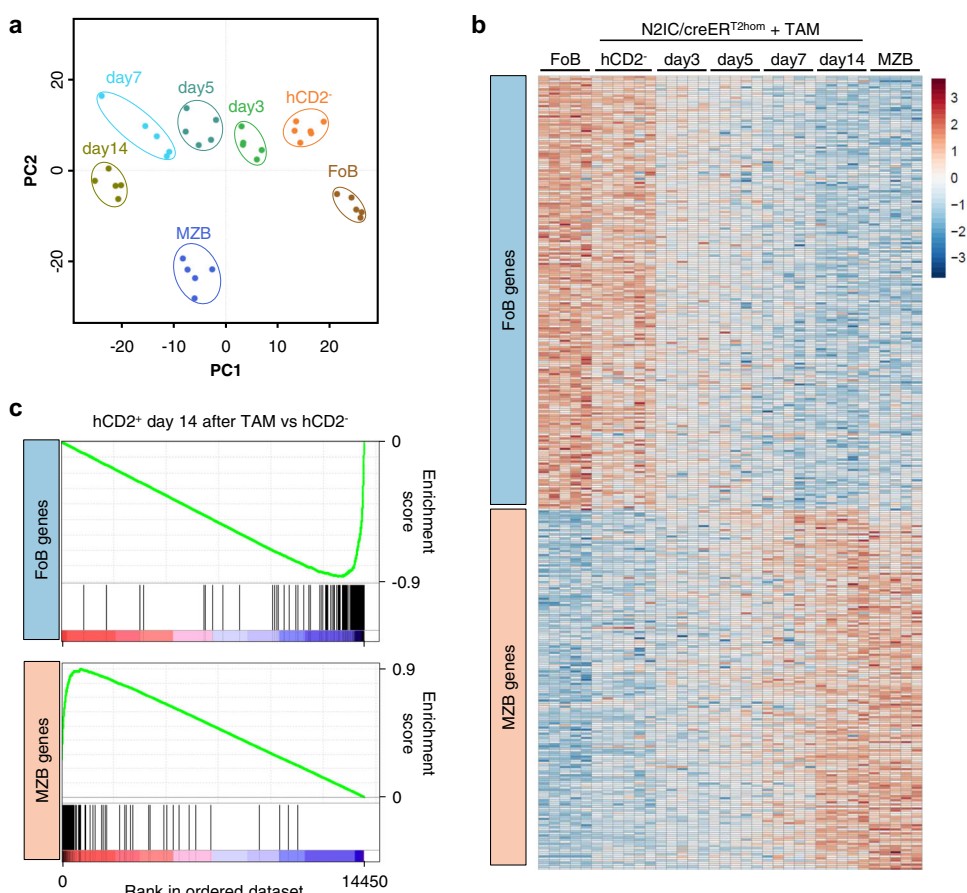

**Fig. 4 Notch2IC induces a transcriptomic shift from FoB to MZB signature.** Results of whole transcriptome RNA sequencing of 5-6 samples each of wild type FoB, MZB and hCD2$^+$ and hCD2$^-$ N2IC/creER$^{T2hom}$ cells 3, 5, 7, and 14 days after tamoxifen (TAM) induction. **a** Unbiased multidimensional scaling (MDS) plot of 10% most variable genes across all sequenced samples. **b** Heatmap of all sequenced samples using predefined MZB signature gene sets ("FoB genes" and "MZB genes") from control FoB and MZB cells as well as Notch2IC-expressing cells over time. Color scheme indicates log2-fold changes normalized across all samples. **c** Preranked GSEA analysis using the FoB and MZB signature gene sets. The ordered dataset displays 14450 genes, ranked for fold changes from upregulated (left) to downregulated (right) in hCD2$^+$ versus hCD2$^-$ cells 14 days after TAM treatment. Enrichment scores are −0.87 for FoB gene set and 0.90 for MZB gene set (NOM $p$-values and FDR $q$-values < 10$^{-3}$ for both).

Notch2-signaling regulates the expression of several G-protein coupled receptors that guide B cells into the MZ and keep them there.

**Mature FoB cells trans-differentiate to MZB cells.** Our data showed that mature FoB cells trans-differentiate to MZB cells upon Notch2IC expression. To investigate whether this trans-differentiation occurs not only in a transgenic setting but also physiologically, we FACS-sorted mature FoB cells from spleens of mice expressing the pan-leukocyte allele variant CD45.1 and transplanted $5 \times 10^6$ highly purified mature FoB cells (B220$^+$ AA4.1$^-$CD23$^+$CD21$^+$) into congenic CD45.2-expressing recipients (Supplementary Fig. 6a). The percentage of recovered CD45.1$^+$ cells within the splenic B cell compartment of host mice was between 0.05% and 0.5% (Supplementary Fig. 6b and c). While on day 1 after transfer, all engrafted CD45.1$^+$ cells still displayed the homogeneous FoB cell phenotype (B220$^+$CD23$^+$ CD21$^+$CD1d$^{low}$) on which the sorting was based, some of the transplanted CD45.1$^+$ cells started to differentiate toward a MZB cell fate (B220$^+$CD21$^{high}$CD23$^{med}$CD1d$^{med}$) at days 4 and 9 (Fig. 6a, b). At day 14 after transplantation a consistent proportion of around 7% of CD45.1$^+$ cells displayed a mature MZB cell phenotype (B220$^+$CD21$^{high}$CD23$^{low}$CD1d$^{high}$) (Fig. 6a–c). These data suggest that highly purified FoB cells can adopt a MZB surface phenotype when transferred into immunocompetent mice.

According to the current knowledge transitional type 2 (T2) (AA4.1$^+$CD21$^+$CD23$^+$) B cells develop stepwise through an intermediate—the so-called MZB cell precursor (MZP) stage (AA4.1$^{low}$IgM$^{high}$CD21$^{high}$CD23$^+$) - to MZB cells (AA4.1$^{low}$ IgM$^{high}$CD21$^{high}$CD23$^{low}$)[9]. To identify and quantify the intermediate populations arising during differentiation of FoB to MZB cells, we applied and further modified the gating strategy published by the Allman group[9]. The initial gating on CD21$^{high}$IgM$^{high}$ B cells (Fig. 7a) contained MZP and MZB cells. Subsequently, mature MZB cells were distinguished from precursors based on their expression of CD23 and CD1d. In controls, around 20% of B cells had a MZP (CD23$^+$, CD1d$^{med}$) and 80% a MZB cell phenotype (CD23$^{low}$CD1d$^{high}$) (Fig. 7b). As expected, a clear CD21$^{high}$IgM$^{high}$ population was missing amongst CD45.1$^+$ cells one day after transfer. However, at days 9 and 14 a subset of the transplanted CD45.1$^+$ FoB cells adopted the phenotype CD21$^{high}$IgM$^{high}$ (Fig. 7a). Subsequent analysis for CD1d and CD23 expression revealed an initial wave of MZP cells, followed by the increase of mature MZB cells (Fig. 7b). These data indicate that the transplanted FoB cells develop first to MZP and then to mature MZB cells through gradual upregulation of CD1d and downregulation of CD23.

The kinetics of CD21, CD23 and CD1d surface marker regulation in transferred CD45.1$^+$ cells (Fig. 6a, b) was in accord with the RNA-Seq analysis of Notch2IC-induced transdifferentiating B cells

a

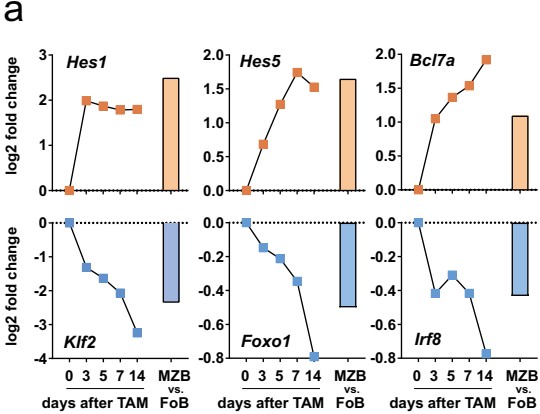

b

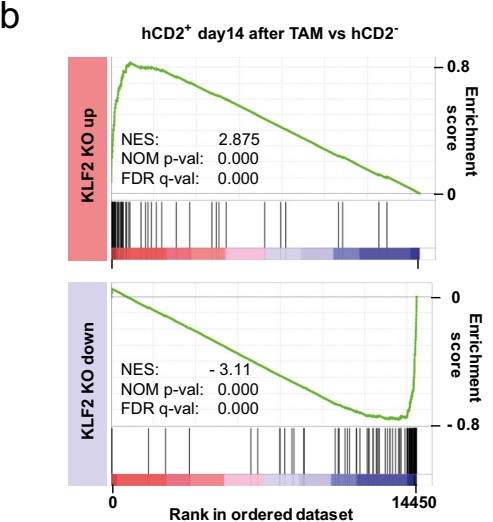

c

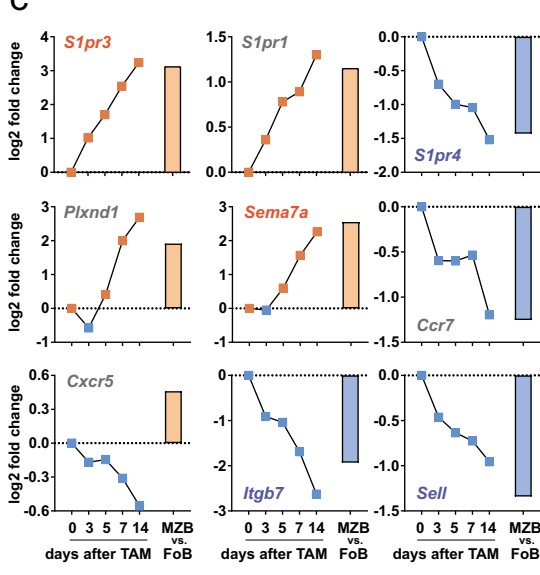

**Fig. 5 Notch2IC regulates genes involved in transcription and homing.**
**a** Mean fold changes in RNA levels of transcription factors and transcriptional regulators in hCD2+ versus hCD2− cells over all time points after TAM treatment. Fold changes of wild type MZB versus FoB samples are shown as bar graph. **b** Enrichment plots of GSEA analysis using Klf2-KO gene sets defined by a microarray analysis of Klf2-deficient FoB cells published by Hart and colleagues[39]. The GSEA shows a significant enrichment of KLF2 target genes at the top and bottom ends in the ranked dataset of hCD2+ (day 14 after TAM) versus hCD2− cells. **c** Mean fold changes in RNA levels of genes involved in B cell migration and MZ homing. Analysis according to (**a**). Colored gene names are KLF2 target genes from (**b**) which are up- (orange) or down- (blue) regulated in the conditional Klf2 knockout.

and CD23 after induction of Notch2IC-expression and transplantation of CD45.1+ FoB cells suggests that the trans-differentiation of FoB to MZB cells is driven by Notch2-signaling in both settings.

## Discussion

Here, we show that Notch2-signaling is sufficient to induce trans-differentiation of FoB to MZB cells, in respect to their surface expression profile, localization and activation status. In the trans-differentiating B cells, the RNA-expression profile gradually changed from a FoB- into a MZB signature within a time window of 14 days. Among the top-regulated genes were several transcriptional regulators, which most likely contribute to the transcriptional changes downstream of Notch2-signaling. The most prominently downregulated transcription factor was KLF2, which is known to restrict the MZB cell compartment. The strong overlap of genes regulated by KLF2 and Notch2 suggests that KLF2 acts as a key regulator of FoB/MZB conversion downstream of Notch2-signaling. Moreover, Notch2-signaling upregulated genes that direct B cells into the MZ and downregulated those that induce homing into the follicle[2,42]. Some of these genes may be indirectly regulated by Notch2-signaling through the downregulation of Klf2, which by itself regulates several genes that are involved in migration[37–39]. Downregulation of Cxcr5 was the only transcriptional regulation caused by Notch2IC that was not in accord with wild type MZB versus FoB differences. Reciprocal receptor activations of CXCR5 and S1PR1 balance the follicular shuttling behavior of MZB cells[2,42]. The constitutive activation of Notch2-signaling may direct all B cells toward the MZ and may prevent their further shuttling. Thus, the low Cxcr5 expression may be a consequence of the enduring Notch2-signaling.

We performed most of our experiments in a CD19-deficient background to have the possibility to follow the generation of newly formed MZB cells in the absence of pre-existing MZB cells. We had previously shown that MZB cells are generated comparably in the presence and absence of CD19 in mice expressing Notch2IC specifically in B cells[21]. These earlier data are in accord with the data presented here, showing that the percentages of MZB cells and the kinetic of MZB cell development is comparable in N2IC/creERT2het and N2IC/creERT2hom mice. These findings exclude that the CD19-deficiency has any effect on the generation of MZB cells after induction of Notch2IC in FoB cells. However, both ongoing Notch2IC-expression and CD19-deficiency might have an effect on the survival of Notch2IC-expressing MZB cells. Hence, the disappearance of hCD2+ B cells over time can not be used to determine the half-life of MZB cells.

After demonstrating that Notch2-signaling was sufficient to drive the trans-differentiation of FoB to MZB cells, we wondered whether this process also occurs physiologically. Indeed, we could show that highly purified FoB cells adopted a MZB cell surface phenotype within a time window of 14 days after transplantation

(Fig. 7c). In both systems the upregulation of CD21 protein and Cr2 mRNA was very fast, while the modulation of CD23 (Fcer2a) and CD1d (Cd1d1) expression was delayed. These data are in line with previous observations suggesting that Cr2 is a direct Notch2 target gene[18,47,48]. In sum, our data show that FoB cells can transdifferentiate into MZB cells in immunocompetent mice within a time window of 14 days. The similar temporal regulation of CD21, CD1d,

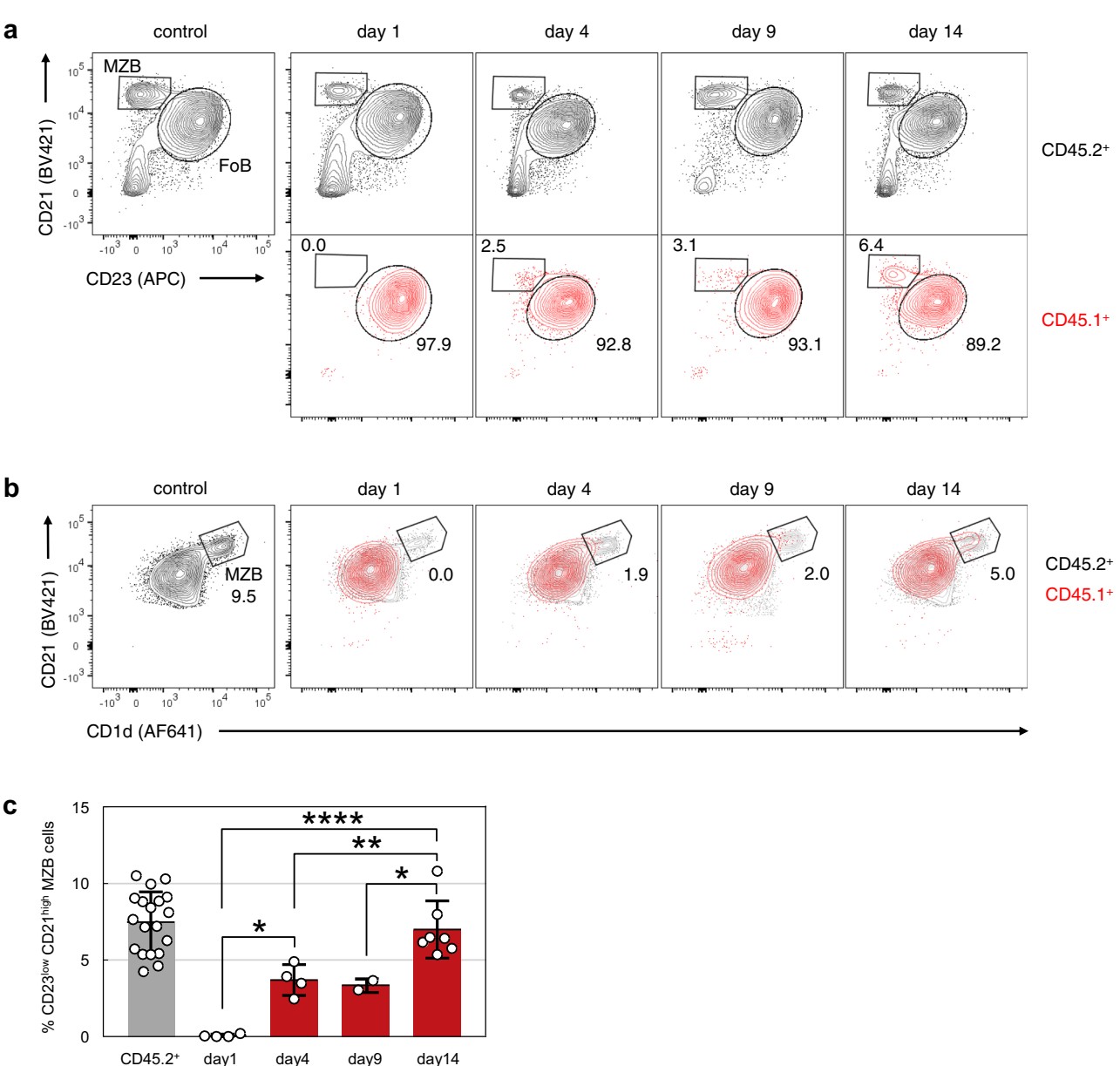

**Fig. 6 FoB cells convert to MZB cells after adoptive transfer.** Flow cytometric analyses of surface phenotypes of recovered CD45.1[+] cells in the spleen of recipient CD45.2[+] mice at the indicated time points after adoptive transfer of purified FoB cells. **a** Representative plots of CD23/CD21 surface phenotype. Cells were gated on B220[+] B cells and then separated for CD45.2[+] recipient cells (black) or corresponding CD45.1[+] recovered donor cells (red). The numbers indicate the percentages of MZB and FoB cells within CD45.1[+] cells. **b** Representative overlays of an CD1d/CD23 surface staining of recovered CD45.1[+] cells (red) and CD45.2[+] recipient cells (gray) at the indicated time points after transplantation. The numbers indicate the percentages of MZB within CD45.1[+] cells. **c** Summary and quantification of percentages of CD23[low]CD21[high] MZB cells among CD45.1[+] cells at different days after transfer. Bar graph shows mean values, SD and individual data points $n = 4$ (d1), $n = 4$ (d4), $n = 2$ (d9), $n = 7$ (d14) independent experiments. (*$p = 0,0111$, ****$p = 1.0E−05$, **$p = 0,0093$, *$p = 0,0249$, one-way ANOVA, Tukey´s multiple comparison test, only CD45.1[+] cells are compared).

into immunocompetent mice. Thus, our data definitively prove that FoB cells can act as precursors for MZB cells in immunocompetent mice. The fact that after transplantation of transitional B cells or B220-deficient BM B cells, MZB cells always appear later than FoB cells would be in agreement with the regular development of MZB cells from FoB cells[7,9]. Some years ago, Cariappa and colleagues discussed the possibility that a recirculating long lived FoB cell population with the phenotype IgM[high]IgD[high]CD21[int]CD23[+], the so-called FoBII cells, act as precursors for MZB cells[28]. Hence, the IgM[high] B cells in our

sorted Fo B cell population might be the precursors for the newly generated MZB cells after transplantation.

It is an interesting question, which FoB cells finally differentiate to MZB cells. Our previous[21] and present data provide evidence that a strong and constitutive Notch2 signal shifts all B cells to the MZB cell compartment. Therefore, we assume that the strength and duration of the Notch2 signal determines whether FoB cells start to differentiate into MZB cells. Liu and colleagues provided evidence that strong Notch2-signaling is initiated in FoB cells rather than T2 cells[49] which further corroborates our conclusion

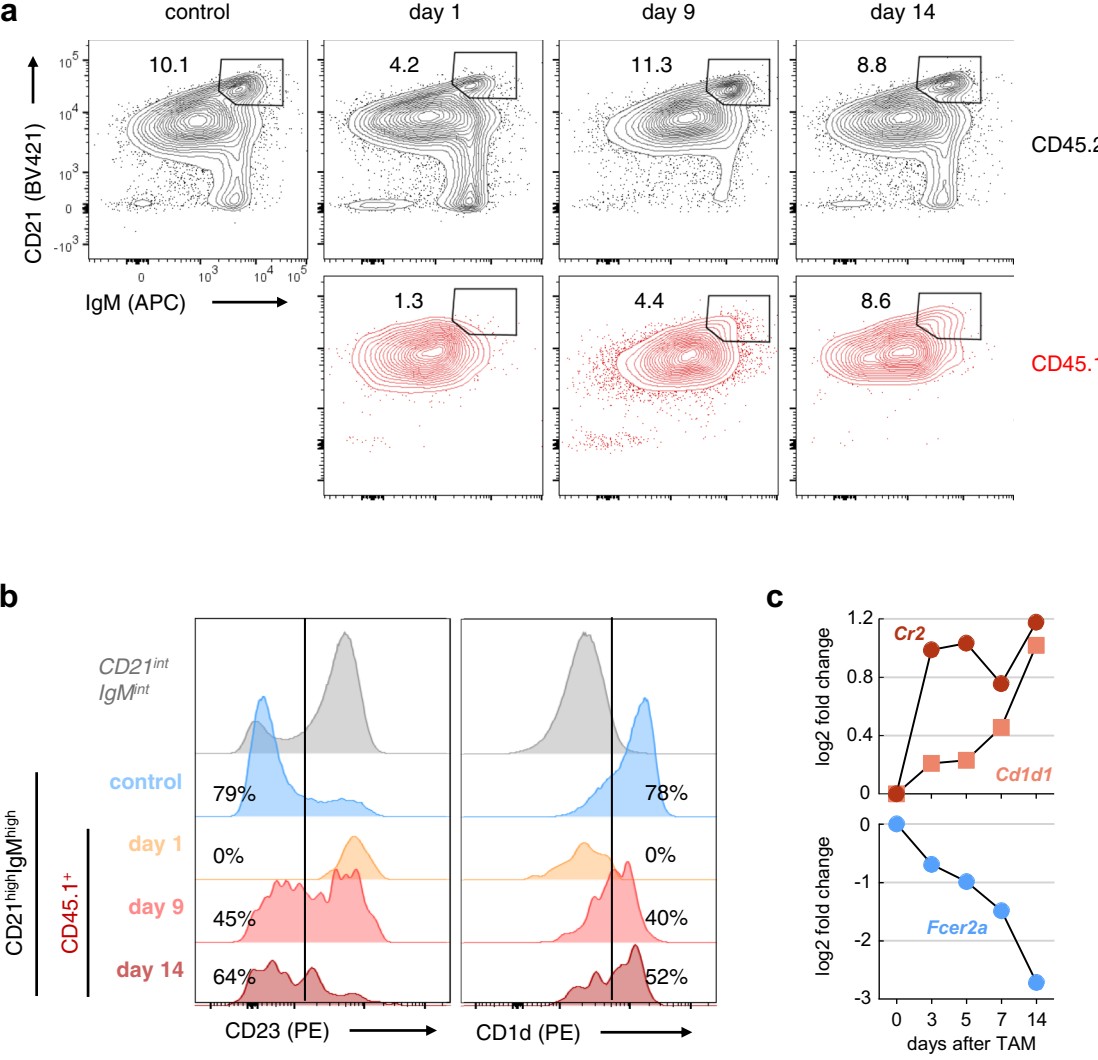

**Fig. 7 FoB cells differentiate to MZB B cells via an MZP cell stage. a** Representative flow cytometric analysis of CD45.2$^+$ recipient and CD45.1$^+$ recovered donor cells for MZB precursor (MZP) and mature MZB cells at the indicated time points after adoptive transfer. Cells were gated for B220$^+$ B cells, followed by identification of CD21$^{high}$IgM$^{high}$ cells, comprising MZP and mature MZB cells. Plots are representative for $n = 4$ (d1), $n = 4$ (d4), $n = 2$ (d9), $n = 7$ (d14) independent experiments. **b** Colored histograms show downstream gating of CD21$^{high}$IgM$^{high}$ for CD23 and CD1d levels to separate CD23$^{low}$CD1d$^{high}$ MZB from CD23$^+$CD1d$^{med}$ MZP cells. The gray histograms represent CD23 and CD1d expression from the CD21$^{int}$IgM$^{int}$ FoB population. **c** Median fold changes in RNA levels of *Cr2* (CD21), *Cd1d* (CD1d) and *Fcer2a* (CD23) in hCD2$^+$ versus hCD2$^-$cells over time after induction of Notch2IC-expression (experiment shown in Fig. 4).

that FoB cells serve as precursors to MZP and MZB cells. The question remains: What are the requirements of FoB cells to receive an above Notch2 threshold signal? Inactivation of one *Notch2* allele in B cells or one *Dll*-1 allele in fibroblasts is sufficient to reduce the percentage of MZB cells compared to wild type mice[18,19], indicating that MZB cell development is highly sensitive to the numbers of ligand/receptor pairs interacting on a single cell. We showed earlier that the avidity of the Notch2/DLL-1 interaction is inherently weak, but increases after glycosylation of Notch2 by the glycosyltransferase Fringe[50]. Accordingly, high expression of a strongly glycosylated Notch2 receptor may be a prerequisite for FoB cells to receive an above threshold Notch2 signal. In addition to Notch2-, also BCR-signaling plays a crucial role in the development of MZB cells: (i) Genetic mutations in BCR-signaling components influence MZB cell development[5,17]. (ii) Some BCR insertions preferentially develop to MZB cells[8,30]. (iii) Self-antigen is necessary for the development of MZB cells[51].

We assume that BCR-signaling acts upstream of Notch2-signaling. Thus, we found that Notch2 surface expression levels are upregulated by BCR-stimulation of FoB cells (Supplementary Fig. 7). Moreover, the group of Bart Lambrecht showed that BCR ligation enhances cleavage of Notch2 by inducing the expression and membrane translocation of the Notch2-processing metalloprotease ADAM10[52]. Consequently, BCR-stimulation seems to enhance both the Notch2-expression on the cell surface and the Notch2-activity. Most interestingly, antigen-presenting follicular dendritic cells (FDCs) are co-localized with DLL1-expressing fibroblasts and B cells in the spleen[23]. This might be the niche where FoB to MZB conversion physiologically occurs, assuming that Notch2-expression and Notch2-signaling are enhanced by simultaneous BCR stimulation.

This DLL-1-expressing niche might be a limiting factor for the development of MZB cells. Several data suggest that B cells compete for DLL-1 expressing cells. (i) Competitive BM transplantation experiments between FringeKO and wild type B cells

revealed that the differentiation of mutant B cells to MZB cells is more strongly decreased in the competitive than in the non-competitive situation, indicating that only B cells with a highly glycosylated Notch2-receptor have the chance to interact with the DLL-1 expressing fibroblasts and to develop to MZB cells[22]. (ii) We show here that after transplantation of wild type FoB cells into wild type mice, the transplanted FoB cells generate physiological percentages of MZB cells, suggesting that the B cells from the donor and recipient mice equally compete for the interaction with DLL-1 expressing fibroblasts to develop to MZ B cells. (iii) In contrast, enhanced MZB cell differentiation has been observed after transplantation of FoB cells in immunodeficient mice and in mice with impaired generation of newly formed B cells as in IL-7KO and RagKO[9,14,15,30,53,54]. This may indicate that a lower competition for the rate limiting DLL1-expressing fibroblasts in lymphopenic mice accelerates and enhances the process of FoB to MZB cell conversion.

Collectively, our data provide strong evidence that FoB cells develop to MZB cells if they receive an above threshold Notch2-signal. We show that Notch2 acts as the key switch for a plethora of phenotypic and transcriptomic alterations, orchestrating the process of in situ identity conversion of FoB to MZB cells. Vice versa, inhibition of Notch2-signaling has been shown to drive MZB cells into the follicle to be indistinguishable from FoB cells[55]. This suggests a significant dynamic plasticity between FoB and MZB cells dependent on Notch2-signaling. Of note, recent work by others provided evidence for a cell-intrinsic lineage plasticity of mature B2 toward B1 cells[56]. Our results finally prove plasticity among mature B cell subsets and suggest that the still widely accepted view of dead-end lineage commitment during B cell development must be broadened.

## Methods

**Mouse models.** Previously described Notch2IC[STOPfl] mice[21] carrying the *Notch2IC[STOPfl]* allele in the *rosa26* locus were crossed with the B cell-specific, tamoxifen-inducible Cre strain CD19-creER[T2,24] to generate either CD19-deficient Notch2IC-STOPfl/wt/CD19-creER[T2hom] (here termed N2IC/creER[T2hom]) or CD19-proficient Notch2IC[STOPfl/wt]/CD19-creER[T2het] mice (here termed N2IC/creER[T2het]). CD19-deficiency was achieved by generation of mice carrying the *creER[T2]* gene in both *Cd19*-loci (creER[T2hom]). R26/CAG-CARΔ1[STOPfl] reporter mice[27] were also crossed to CD19-creER[T2] mice as controls (here termed CAR/creER[T2hom]). In addition, CD19-proficient CD19-creER[T2het] mice were used as controls. All above-mentioned strains are of mixed C57BL/6J × BALB/c background. "Wild type" control MZB and FoB cells for RNA Seq experiments were purified from C57BL/6J × BALB/c F1 mice. Male and female mice at the age of 10–20 weeks were used. Only females were used for RNA sequencing experiments. C57BL/6J animals with the leukocyte marker CD45.1 (Ly5.1) were used for adoptive transfer experiments into sex-matched congenic CD45.2 C57BL6/J recipients CD45.1 mice. C57BL/6J CD45.1 (B6.SJL-Ptprc[a] Pepc[b]/BoyJ) and C57BL/6J CD45.2 were purchased from the Jackson Laboratories and bred and maintained in house. Animals were bred and maintained in specific pathogen-free conditions and experiments were performed in compliance with the German Animal Welfare Law and were approved by the Institutional Committee on Animal Experimentation and the government of Upper Bavaria.

**Mouse treatments and immunizations.** Cre activity was induced by oral gavage of 5 mg tamoxifen, dissolved in 200 µl corn oil (Sigma) in a single dose (day 1) or in 5 dosages applied on 5 consecutive days (day 1–5). In all single-dose experiments, mice were pre-treated with four doses of 250 µg (250 µl) anti-IL7R antibody (A7R34) administered intraperitoneally (i.p.) every second day. To trigger T-cell independent (TI) immune responses, mice were immunized i.p. with 10 µg NP-LPS (LGC Biosearch) in 200 µl sterile PBS.

**Flow cytometry.** Single cell suspensions were generated from spleen and bone marrow. Surface stainings of lymphocytes were performed on ice for 20 min in MACS buffer (Miltenyi). For intracellular FACS-stainings, cells were fixed in 2% formaldehyde (1:2 PBS diluted Histofix) for 5 min at room temperature and permeabilized in ice-cold 100% methanol for 10 min on ice. Cells were stained for 45 min at room temperature with the corresponding antibodies. Cytometric analysis was performed on a FACSCalibur or LSRFortessa (BD Biosciences). Results were analyzed and visualized with FlowJo V10. General gating strategies are indicated in Supplementary Fig. 8.

**Histology.** Splenic tissues were embedded in O.C.T., snap frozen and stored at −20 °C. 7 µm sections were mounted on glass slides and stored at −80 °C for immunohistochemical (IHC) analysis.

For chromogenic IHC, slides were fixed with ice-cold acetone for 10 min, washed in PBS, blocked for 20 min in PBS supplemented with 1% BSA, 5% goat serum. Subsequently, sections were blocked using the Avidin/Biotin blocking kit (Vector) according to the manufacturer's protocol. Expression of hCD2 was detected immunohistochemically by staining with biotinylated mouse anti-human CD2 antibody and rabbit anti-laminin antibody at 4 °C overnight in 1% BSA/PBS. Secondary staining was performed for 1 h at room temperature using streptavidin-coupled alkaline phosphatase and peroxidase-coupled anti-rabbit IgG. AEC substrate kit and Blue AP substrate kit (Vector) were used according to the manufacturer's protocol for final enzymatic chromogenic detection. Slides were embedded in Kaiser's Gelatine (Carl Roth) and analyzed using an Axioscope (Zeiss) equipped with an Axiocam MRc5.

For immunofluorescent (IF) staining, tissue slides were fixed with 3% Histofix (Carl Roth, diluted in PBS) for 10 min, rinsed with PBS, rehydrated for 5 min in PBS + 50 mM NH₄Cl and then blocked in PBS supplemented with 1% BSA, 5% rat serum, 5% chicken serum for 30 min followed by Avidin/Biotin blocking (Vector) according to the manufacturer's protocol. Primary (goat anti-mouse IgM, anti-Thy1.2-Biotin), secondary (chicken anti-goat IgG AF647) and fluorophore-coupled antibodies (anti-IgD-FITC, anti-MOMA-FITC), as well as Streptavidin-AF594, were incubated for 1 h at room temperature in 0.5% BSA/PBS. Slides were mounted in ProLong Glass Antifade (Invitrogen). IF images were acquired on a TCS SP5 II confocal microscope (Leica). Picture stacks were composed using ImageJ.

**In vitro assays.** For plasmablast-differentiation assays, naïve B cells were isolated from splenic cell suspension by using the CD43 depletion B cell Isolation Kit (Miltenyi) according to the manufacturer's instructions. To monitor cell division B cells were stained with CellTrace Carboxyfluorescein succinimidyl ester (CFSE) (Invitrogen) according to the manufacturer's protocol. Cells were cultured for 48 h at a density of $5 \times 10^5$ cells/ml in flat-bottom 96 well plates in RPMI-1640 (Gibco) supplemented with 10% fetal calf serum (FCS), 1% L-glutamine, 1% non-essential amino acids, 1% sodium pyruvate, 50 µM β-mercaptoethanol and stimulated with 50 µg/ml lipopolysaccharide (LPS, Sigma). Cells were harvested after 48 h and analyzed by flow cytometry after staining with anti-hCD2-APC and anti-CD138-PE antibodies. Dead cells were discriminated from living cells using a fixable dead cell staining kit (Invitrogen).

For Notch2 surface expression analysis, naïve FoB cells of control mice were purified from splenic cell suspension using the MZ and Fo B Cell Isolation Kit (Miltenyi). $5 \times 10^5$ FoB cells/well were cultured as described above but stimulated with 15 µg/ml α-IgM for up to 72 h until harvest and staining with anti-Notch2-PE antibody.

**Cell sorting and RNA purification.** Prior to cell sorting, mice were pretreated with anti-IL7R antibody and Notch2IC-expression was induced by application of a single dose of 5 mg tamoxifen (Sigma) /200 µl corn oil (Sigma) by oral gavage. Per time point splenic hCD2⁺B220⁺ cells were sorted from $n = 5$ mice. As controls the following cell populations were sorted: hCD2⁻B220⁺ B cells from tamoxifen-treated mice of different time points ($n = 6$ samples) as well as FoB (B220⁺AA4.1⁻CD23[high]CD21⁺) and MZB (B220⁺AA4.1⁻CD23[low]CD21[high]) cells from 5 untreated C57BL/6J × BALB/c F1 mice. Cell sorting was performed with a FACSAria III (BD). For every sample, $5 \times 10^4$ cells were directly sorted into 1 ml of cooled (4 °C) TRIzol LS (Invitrogen), snap frozen on dry ice and stored at −80 °C. RNA was purified using a quick hybrid TRIzol/column-wash protocol. Briefly, after thawing of TRIzol samples, 200 µl chloroform was added, and RNA separated into the aqueous phase via centrifugation at $12,000 \times g$, 4 °C for 15 min. The RNA phase was transferred into a reaction tube and mixed with an equal amount of 100% EtOH. Samples were subsequently loaded on ReliaPrep RNA Cell Miniprep columns (Promega) and further purified according to the supplied protocol. RNA purity was confirmed using RNA Pico chips on a 2100 Bioanalyzer (Agilent). Highly pure RNA with mean RNA integrity number (RIN) of 9.7 (SD = 0.4) was used for subsequent library preparation and RNA sequencing.

**RNA sequencing.** Library preparation for bulk 3′-sequencing of poly(A)-RNA was done as described previously[57]. Briefly, barcoded cDNA of each sample was generated with a Maxima RT polymerase (Thermo Fisher) using oligo-dT primer containing barcodes, unique molecular identifiers (UMIs) and an adapter. 5′ ends of the cDNAs were extended by a template switch oligo (TSO) and after pooling of all samples full-length cDNA was amplified with primers binding to the TSO-site and the adapter. cDNA was tagmented with the Nextera XT kit (Illumina) and 3′-end-fragments finally amplified using primers with Illumina P5 and P7 overhangs. In comparison to Parekh et al.[57], the P5 and P7 sites were exchanged to allow sequencing of the cDNA in read1 and barcodes and UMIs in read2 to achieve a better cluster recognition. The library was sequenced on a NextSeq 500 (Illumina) with 65 cycles for the cDNA in read1 and 16 cycles for the barcodes and UMIs in read2.

Data were processed using the published Drop-seq pipeline (v1.0) to generate sample- and gene-wise UMI tables. Reference genome (GRCm38) was used for

alignment. Transcript and gene definitions were used according to the ENSEMBL annotation release 75.

**Bioinformatics**. The murine reference genome GRCm38 was used as reference for mapping the raw read data with Dropseq tools v1.13[58]. Gencode gene annotation release M19 was used to determine read counts per gene. The resulting genes x samples count matrix was imported into R v3.4.4 and further processed with DESeq2 v1.8[59]. Lowly expressed genes, e.g., genes having in sum less than 10 reads across all samples, were removed prior to any downstream analysis. The covariates phenotype and time point were combined to a dummy covariate describing all possible time point phenotype combinations. Dispersion of the data was determined with a parametric fit, including the described dummy variable in the design. The 10% most variant expressed genes across all samples based on the rlog transformed expression values were selected and used as input for multidimensional scaling (MDS). Differential expression between gradual time points and MZB versus FoB cells was determined with a Wald test. For the generation of the MZB signature gene sets, the gene list for differentially regulated genes between wild type MZB and FoB cells were downloaded from the ImmGen Population Comparison tool[34] (microarray data GSE11961[60]) and intersected with the respective comparison in this dataset. All Gene Set Enrichment Analyses (GSEA) were performed with GSEA v4.0.1 or v4.0.2 from Broad Institute. For GSEA the log2 fold change was used as ranking metric for genes.

**Adoptive transfer experiments**. FoB cells (B220$^+$AA4.1$^-$CD23$^{high}$CD21$^{mid}$) were FACS-sorted from splenocytes of CD45.1$^+$ donor animals. $5 \times 10^6$ highly purified FoB cells were washed and resuspended in 200 μl sterile PBS and injected into the lateral tail vein of congenic CD45.2$^+$ recipient animals. Splenocytes of recipient mice were analyzed at days 1, 4, 9, and 14 by flow cytometry for the surface phenotype of engrafted CD45.1$^+$ B cells over time. 2–5 million cells were recorded on a LSRFortessa to ensure a faultless representation of at least 10$^3$ CD45.1$^+$ cells in contour plots.

**Statistics**. *p*-values of indicated one-way or two-way ANOVA with multiple comparison tests were determined using GraphPad Prism8. The p-value for correlation analysis of fold-changes between MZB/FoB and hCD2+/hCD2− sample pairs as shown in Supplementary Fig. 5 was executed in MS Excel using the "Analysis ToolPak" Add-In.

**Reporting summary**. Further information on research design is available in the Nature Research Reporting Summary linked to this article.

## Data availability

RNA sequencing data used and analyzed in this study have been deposited in ENA under the accession code PRJEB35207. All other data are available from the corresponding author upon reasonable request. Source data are provided with this paper.

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

## Acknowledgements

This work was supported by the Deutsche Forschungsgemeinschaft (DFG ZI1382/4-1 and DFG SCHM2440/6-1; SFB1243, projects A12 and A13). We thank the animal facility of the Helmholtz Center and our animal care takers for excellent housing of the mice and Krisztina Zeller for help with genotyping. Our gratitude goes to Rupert Öllinger for performing the RNA sequencing reactions. We thank the Heissmeyer lab for providing CD45.1 mice and Annette Frank from the lab of Irmela Jeremias for her precious help with intravenous injection for adoptive transfers.

## Author contributions

M.L. designed, performed and analyzed most experiments. T.E. analyzed RNA sequencing data. L.J.S. performed additional computational analysis. T.B. performed the in vitro BCR-stimulation experiment and analyzed BrdU data. R.R. provided cooperation for RNA-sequencing and bioinformatic analysis. M.S.-S. contributed to experimental design and manuscript preparation. M.L. and U.Z.-S. wrote the manuscript with contribution of all authors. L.J.S. and U.Z.-S. contributed equally. All authors read and approved the final paper.

## Funding

## Competing interests

The authors declare no competing interests.
