## [Peer Review File · Nature Communications]

Editorial note: Parts of this Peer Review File have been redacted as indicated to remove third-party material where no permission to publish could be obtained.

REVIEWER COMMENTS

Reviewer #1 (Remarks to the Author):

This manuscript convincingly demonstrates that turning on Notch in follicular B cells can lead to marginal zone B cell differentiation. The studies are elegant and very cleanly executed. The results are however consistent with an earlier view that FO2 B cells (IgDhi IgMhiCD21 intermediate) may serve as a reservoir for the generation of marginal zone B cells. Perhaps adding this to the discussion may be appropriate.

Reviewer #2 (Remarks to the Author):

Previous work has established a critical role for Notch2 and Delta like ligand-1 (DLL1) in MZ B cell development in the spleen. While the general paradigm that has been established is that transitional B cells give rise to MZ B cells in a Notch2 dependent manner, some work has also shown that mature recirculating B cells can give rise to MZ B cells. For example, a study in the rat showed that B cells from thoracic duct lymph (hence lymph node derived cells that are mature, follicular cells) can give rise to MZ B cells (Kumaratne and MacLennan EJI 1981 11:865). A study by You/Carter (JI 2009 182:7343) showed the ability of wild-type splenic B cells to give rise to MZ B cells after transfer into CD19-deficient hosts.

The present study uses a mouse strain that allows inducible activation of Notch2 in B cells in mice that lack endogenous MZ B cells (CD19 KO). Activation of Notch2 signaling in B cells in mice where B cell development had been blocked led to the appearance of MZ B cells. These cells were shown to be very similar to normal MZ B cells in terms of their ability to differentiate into plasma cells in response to LPS, and by an extensive gene expression comparative analysis. Finally, sort purified mature follicular B cells are shown to give rise to MZ B cells and this appears to occur through a MZ precursor phenotype step. Overall, the experiments are well performed and the results provide definitive data that Notch2 pathway activity is sufficient to promote follicular B cell differentiation into MZ B cells. However, this finding is considered an incremental advance over previous work. The study does not challenge the view that many MZ B cells develop from transitional B cells. It just strengthens the evidence that follicular B cells can give rise to MZ B cells.

Other comments:

1. The 'CAR' acronym in the results is not well introduced and it took me sometime to know what it meant.
2. The first sentence in the discussion states 'the RNA-expression profile gradually changed from a MZB- into a FoB signature' after activating Notch2 signaling. I think the reverse is meant.
3. A more comprehensive assessment of the past literature is needed and the work needs to be put more accurately into context of the full body of past work.
4. It remains unclear how the size of the MZ B cell population is determined. The suggestion that the number of DLL1 expressing stromal cells in the follicle is limited and this controls the number of MZ B cells that can be generated is found to be very vague and doesn't represent an advance over our previous understanding from the work in ref 16 showing the critical role of CCL19-expressing stromal cells as a DLL1 source.

Reviewer #3 (Remarks to the Author):

The manuscript by Lechner and Zimmer-Strobl and their colleagues describes experiments that provide compelling evidence for the existence of lineage plasticity between follicular B cells (FoB) and marginal zone B cells (MZB) that is mediated by the NOTCH2 transcription factor. NOTCH2

activation is demonstrated to reprogram FoB into MZB not only regarding their phenotype, but also their function. Interestingly, in the discussion the authors provide a comprehensible hypothesis as to how one can envisage that MZB cells, that clearly are reactive against certain antigens, can form from FoBs following depletion of MZB in the host.

The manuscript contains high-quality data throughout with excellent presentation of the results, and the IF shown in Fig. 2 is outstanding. The inclusion of an RNA-seq analysis in Fig. 4 dissipates any doubt that we are not looking at MZB generated gradually from FoB by induction of NOTCH2 expression. However, as with many of such studies that require intricate transgenic approaches, there are a couple of issues that require attention.

1) Several key experiments are performed on a CD19-deficient background, one reason for this being that no MZB are generated in Cd19-deficient mice. Could the CD19-deficiency have influenced the results of the study and their interpretation in any way? It probably did not, but I think this warrants a paragraph in the discussion. That CD19-deficiency does do something is evident from page 10 where it is stated that "neither wild type FoB and hCD2-minus cells nor wild type and trans-differentiated MZB cell clustered together", which is explained by "this basic offset is most likely due to the CD19-deficiency of N2IC/creER-T2hom mice in comparison to wild type mice".

2) On page 5, it is explained that "...the proportion of hCD2+ B cells increased over time most likely due to their proliferation...". Why would they proliferate? Is this expected from known B-cell biology? This should be commented.

3) Page 7: Is it an established fact that only MZB cells among isolated B cells react to LPS in vitro with plasmablast development? If so, perhaps this knowledge should be stated in a half sentence.

4) Reg. the adoptive transfer experiments with FoB cells in the immunocompetent mice (Fig. 6): why exactly can it be excluded that the MZB do not develop from a very minor contaminating MZB population? Especially since there seems to be some proliferation involved (see criticism #1 above).

Minor:

1) On page 5, I suggest to exchange "...continuously approached..." (this expression seems somewhat convoluted) by "...gradually acquired..." a MZB cell surface phenotype.

2) On page 6, one should probably add after "...induces a phenotypic conversion from FoB towards MZB cell..." "..., at least with regard to the markers analyzed, ...". The RNA-seq analysis which justifies this sentence comes only later.

Comments to reviewers #1–3

We are grateful for the valuable suggestions, comments and questions of the reviewers, which we tried to answer satisfactorily. We are convinced that we could address most of the reviewers' comments by extending our introduction and discussion. We believe that the changes we made following the reviewers' suggestions have substantially improved the quality of our manuscript.

Answers to the comment of reviewer #1:

This manuscript convincingly demonstrates that turning on Notch in follicular B cells can lead to marginal zone B cell differentiation. The studies are elegant and very cleanly executed. The results are however consistent with an earlier view that FO2 B cells ($IgD^{hi} IgM^{hi} CD21^{intermediate}$) may serve as a reservoir for the generation of marginal zone B cells. Perhaps adding this to the discussion may be appropriate.

Cariappa and colleagues described in 2007 in JI the existence of two functionally distinct long-lived B cell populations, the so-called FoBI and FoBII cells. In contrast to MZP cells, FoBII cells develop independently of Notch2. In their discussion, the authors suggested that these cells might serve as precursors for MZB cells. We agree with reviewer #1 that this point has to be included in our discussion.

We have now extended our discussion to include the following: *Some years ago, Cariappa and colleagues discussed the possibility that a recirculating long-lived FoB cell population with the phenotype $IgM^{high} IgD^{high} CD21^{int} CD23^{+}$, the so-called FoBII cells, act as precursors for MZB cells (Cariappa et al., JI, 2007). Thus, the IgM^{high} B cells in our sorted Fo B cell population might be the precursors for the newly generated MZB cells after transplantation.*

Answers to the comments of reviewer #2:

1. Previous work has established a critical role for Notch2 and Delta like ligand-1 (DLL1) in MZ B cell development in the spleen. While the general paradigm that has been established is that transitional B cells give rise to MZ B cells in a Notch2 dependent manner, some work has also shown that mature recirculating B cells can give rise to MZ B cells. For example, a study in the rat showed that B cells from thoracic duct lymph (hence lymph node derived cells that are mature, follicular cells) can give rise to MZ B cells (Kumararatne and MacLennan EJI 1981 11:865). A study by You/Carter (JI 2009 182:7343) showed the ability of wild-type splenic B cells to give rise to MZ B cells after transfer into CD19-deficient hosts.

We agree with reviewer #2 that some earlier studies hinted to a role of FoB cells as precursor's for MZB cells. However, to our knowledge up to now it has not been definitively proven nor is it generally accepted knowledge that FoB cells really act as precursors for MZB cells in an immunocompetent setting. Thus, the differentiation of MZB cells in mice and rat may be different. In particular, because unlike in mice, MZB cells are the major mature B cell population in rats (Kumararatne et al. EJI, 1981). Development of MZB cells after adoptive transfer of FoB cells was observed in immunodeficient mice (Agenes and Freitas et al., JEM, 1999; Vinuesa et al., EJI 2003; Srivastava et al., JEM, 2005). It was suggested that in lymphopenic mice, FoB cells are under immediate antigenic pressure resulting in their aberrant activation accompanied by a MZ-like phenotype and PC differentiation (Martin and Kearney, Nat Reviews Immunol., 2002). In contrast, MZB cell development was not observed after transfer of FoB cell into immunocompetent mice (Srivastava et al., JEM, 2005). The difference to our study is probably due to the time window, in which the transplanted mice were analyzed. Since we knew from our Notch2IC induction kinetics that the trans-differentiation from FoB to MZB cells takes quite long, we analyzed the mice up to 14 days after transplantation and showed that this period is necessary to finally complete MZB cell differentiation.

We have extended our introduction according to the suggestions of reviewer #2.

- We mention and reference the original findings which led to the current model that in mice MZB cells develop from transitional B cells (Loder et al., JEM, 1999; Martin and Kearney, Immunity, 2000; Srivastava et al., JEM, 2005)
- We describe the observation that FoB cells are the precursors of MZB cells in rats (Kumararatne et al. EJI, 1981; Dammers et al., EJI, 1999).
- We describe that MZB cell differentiation was observed after transplantation of FoB cells in immunodeficient but not in immunocompetent mice (Agenes and Freitas et al., JEM, 1999; Vinuesa et al., EJI, 2003; Srivastava et al., JEM, 2005).
- We did not include the study of You et al., JI, 2009 describing that the adoptive transfer of splenic wildtype B cells gives rise to MZB cells after transplantation in CD19-deficient mice. Since the transferred splenic B cell population contained most likely transitional B cells, it cannot be discriminated whether the MZB cells were generated from transitional or FoB cells.

2. The study does not challenge the view that many MZ B cells develop from transitional B cells. It just strengthens the evidence that follicular B cells can give rise to MZ B cells.

We agree with the reviewer that although we show that MZB cells physiologically can originate from FoB cells we cannot draw the conclusion that most MZB cells are generated from FoB cells. The main source of newly generated MZB cells could also vary depending on the age of the mice and the proportion of de novo B cell production from the bone

marrow. In our revised manuscript, we discuss the indications that could support the development of MZB cells from FoB rather than transitional B cells:

- After transplantation of transitional B cells or B220-deficient BM B cells, MZB cells appear always later than FoB cells (Loder et al., JEM, 1999; Martin and Kearney, Immunity, 2000; Srivastava et al., JEM, 2005). Therefore, it cannot be excluded that MZB cells originate from the earlier generated FoB cells.
- We show that within 14 days after transplantation of FoB cells the physiological MZB/FoB cell ratio is formed, suggesting that FoB cells can efficiently act as precursors for MZB cells.
- We assume that B cells, which receive an above threshold Notch2 signal are the precursors for MZB cells (see below, point 6). Recently, it has been shown that strong Notch2-signaling is initiated in FoB cells rather than T2 cells, which further corroborates our conclusion that FoB cells serve as precursors to MZP and MZB cells. (Liu et al., MCB, 2015)
- Some years ago, Cariappa and colleagues discussed the possibility that a recirculating long lived FoB cell population with the phenotype $IgM^{high}IgD^{high}CD21^{int}CD23^{+}$, the so-called FoBII cells, act as precursors for MZB cells (Cariappa et al., JI, 2007). Our transplantation experiments are in accord with this hypothesis.

In summary, we do not claim that most MZB cells develop from FoB cells, but our data highlight FoB cells as an additional meaningful source of MZB development.

3. The 'CAR' acronym in the results is not well introduced and it took me sometime to know what it meant.

We apologize for this error and have now included an explanation:

As controls, we used reporter mice expressing a truncated version of the human coxsackie/adenovirus receptor (CAR) upon Cre mediated recombination (R26/CAG-CARD1^{StopF}). The CAR receptor can be stained at the cell surface by FACS.

4. The first sentence in the discussion states 'the RNA-expression profile gradually changed from a MZB- into a FoB signature' after activating Notch2 signaling. I think the reverse is meant.

We thank the reviewer for alerting us to this mistake and have changed it in the revised manuscript.

5. A more comprehensive assessment of the past literature is needed and the work needs to be put more accurately into context of the full body of past work.

- We collected the arguments, which point to the development of MZB cells either from transitional or from FoB cells. We inserted these points into the introduction. We additionally cite in the introduction the work of Loder et al., JEM, 1999; Martin and Kearny, Immunity, 2000 and Srivastava et al., JEM, 2005, who showed that transitional B cells give rise to MZB cells after transplantation in Rag-/- mice. As indications for the development of MZB cells from FoB cells we mention that (i) FoB cells act as precursors for MZB cells in rats (Kumaratne et al., EJI 1981a; Dammers et al., EJI, 1999), (ii) FoB cells give rise to MZB cells after transplantation in immunodeficient mice (Agenes and Freitas, JEM, 1999; Vinuesa et al., EJI, 2003; Srivastava et al., JEM, 2005). We discussed these points in more detail in the answer to question 1.
- We included in the discussion that the MZB cells, which were generated after transplantation of FoB cells into immunocompetent mice, might arise from the recently described FoBII cells with the phenotype $IgM^{high}IgD^{high}CD21^{int}$ (Cariappa et al., JI, 2007).
- We inserted in the introduction an additional paragraph describing the current knowledge about the signaling pathways that are necessary for the development of MZB cells. Here we cite for an overview the reviews of Lopes-Carvalho and Kearney, Immunological Review, 2004; Pillai et al., Annu Rev Immunol., 2005; Pillai and Cariappa, Nature Reviews Immunol., 2009. In this context, we inserted some sentences about the role of the BCR in the development of MZB cells.

Considering the aforementioned knowledge, we defined the following open questions, which we wanted to answer in our study:

1. Can FoB cells act as precursors for MZB cell in immunocompetent mice?
2. Is Notch2-signaling sufficient to induce differentiation of FoB to MZB cells?

6. It remains unclear how the size of the MZ B cell population is determined. The suggestion that the number of DLL1 expressing stromal cells in the follicle is limited and this controls the number of MZ B cells that can be generated is found to be very vague and doesn't represent an advance over our previous understanding from the work in ref 16 showing the critical role of CCL19-expressing stromal cells as a DLL1 source.

We agree with the reviewer that regulation of the MZB cell compartment size is a very interesting but also complex issue, which has been discussed in the field for a long time. In the present manuscript, we had the intention to address the trans-differentiation from FoB to MZB cells and to answer the question whether Notch2-signaling is sufficient for this process. Thus, in our opinion addressing how the size of the MZB cell compartment

is regulated is out of scope of our manuscript. However, our data showing that Notch2 signaling is sufficient to induce trans-differentiation of FoB cells to MZBs allowed us to make some assumptions in the discussion regarding the MZB cell compartment size.

We assume that the Notch2-signaling strength is the key player in regulating the size of the MZB cell compartment. We suggest that the MZB cell compartment size is limited, because only a few B cells have the prerequisite to receive an above threshold Notch2-signal and therefore can develop to MZB cells. The Notch2-signaling strength can be regulated on different levels, such as Notch2-surface expression, glycosylation of the Notch2-receptor, as well as by the expression level of the Notch2-processing protease Adam 10. These parameters might be partially regulated by BCR-signaling. Some data indicate that B cells have to compete for a rate limiting number of DLL-1-expressing fibroblasts. Therefore, we assume that only B cells that express high levels of strongly glycosylated Notch2 receptors interact with the DLL-1-expressing fibroblasts for the time needed to receive an above threshold signal. There are some indications for this hypothesis. (i) Competitive BM transplantation experiments between FringeKO and wt B cells revealed that the differentiation of mutant B cells to MZB cells is more strongly decreased in the competitive than in the non-competitive situation (Tan et al., *Immunity*, 2009). (ii) The emergence of a physiological ratio of MZB/FoB cells after transplantation of FoB cells in immunocompetent wildtype mice (our present study) suggests that B cells from the donor and recipient mice have the same chance to interact with DLL-1 expressing fibroblasts and to develop to MZB cells. (iii) After transplantation of FoB cells in immunodeficient mice (Agenes and Freitas, *JEM*, 1999, Vinuesa et. al., *EJI*, 2003, Srivastava et al., *JEM*, 2005) and in mice with impaired generation of newly formed B cells (Hao and Rajewsky, *JEM*, 2001, Carvalho et al., *JEM*, 2001), MZB cell differentiation is enhanced. This may be due to the low B cell numbers, which increases the chance for the interaction of FoB cells with DLL-1 expressing fibroblasts. (iv) Haploinsufficiency of Notch2 or Dll1, resulting in strongly reduced MZB cell percentages, points out that the halving of the receptor or ligand density is already sufficient that hardly any B cell reaches the necessary Notch signal threshold levels for MZB differentiation (Saito et al., *Immunity*, 2003; Hozumi et al., *Nat. Immunol*, 2004).

We have included these arguments in the discussion.

Answers to the comments of reviewer #3:

1. Several key experiments are performed on a CD19-deficient background, one reason for this being that no MZB are generated in Cd19-deficient mice. Could the CD19-deficiency have influenced the results of the study and their interpretation in any way? It probably did not, but I think this warrants a paragraph in the discussion. That CD19-deficiency does do something is evident from page 10 where it is stated that “neither wild type FoB and

hCD2-minus cells nor wild type and trans-differentiated MZB cell clustered together”, which is explained by “this basic offset is most likely due to the CD19-deficiency of N2IC/creER-T2^{hom} mice in comparison to wild type mice”.

We have included an additional paragraph in the discussion, in which we address the influence of the CD19-deficiency on our experiments. Comparison of the MZB cell differentiation in N2IC/creERT2^{het}- and N2IC/creERT2^{hom}- mice, which have one or no functional CD19-allele, respectively, revealed a comparable MZB cell development (see also Figure 1d+e (CD19-deficient) as well as Supplementary Figure 2c+d (CD19-proficient)). Therefore, we are convinced that the CD19-deficiency does not have any effect on the trans-differentiation of Notch2IC-expressing FoB cells into MZB cells. However, we cannot exclude that the CD19-deficiency has an influence on phenotypes, which we do not describe in the manuscript. Thus, the CD19-deficiency may impair the survival of MZB cells. Hence, the disappearance of hCD2⁺ cells over time cannot be used to calculate the half-life of MZB cells.

2. On page 5, it is explained that “...the proportion of hCD2+ B cells increased over time most likely due to their proliferation...”. Why would they proliferate? Is this expected from known B-cell biology?

MZB cells have a self-renewal capacity (Hao and Rajewsky, JEM, 2001) and thus proliferate to a certain extent. This is in accord with earlier data showing a higher BrdU incorporation in MZB cells in comparison to FoB cells (for example described in Cariappa et al, 2007; Srivastava et al, 2005).

[Redacted]

During in vivo pulse chase BrdU-experiments, mainly the rapidly dividing developing B cells in the BM incorporate BrdU. To rule out that the BrdU labelling of the FoB and MZB cells was caused by their new formation from immature cells during the BrdU pulse, we determined the BrdU incorporation in the presence of anti-IL-7R treatment. Notch2IC/creER^{T2het} mice were treated 4 times with anti-IL-7R antibody before Notch2IC expression was induced by administration of Tamoxifen (d0). After tamoxifen administration mice were fed with BrdU for 7d. Subsequently, the BrdU-incorporation was determined in hCD2⁻ FoB and MZB cells as well as in hCD2⁺ Notch2IC expressing cells.

We found that: (i) Both hCD2⁻ MZB and hCD2⁺ cells incorporate more BrdU than hCD2⁻ FoB cells. (ii) The percentages of BrdU⁺ cells were similar in wt MZB cells and hCD2⁺ B cells. (iii) In both MZB and hCD2⁺ cells, only a certain fraction of B cells incorporated BrdU. Since the BrdU-incorporation was comparable in hCD2⁺ and wt MZB cells we concluded that the increased percentage of BrdU⁺ cells in hCD2⁺ B cells in comparison to hCD2⁻ and CAR⁺ cells (see also Fig. 1g) is due to the acquisition of a MZB cell phenotype.

We have included these new results in Figure 1h.

Page 7: Is it an established fact that only MZB cells among isolated B cells react to LPS in vitro with plasmablast development? If so, perhaps this knowledge should be stated in a half sentence.

We have added the following text in the results section:

- *MZB cells respond much faster than FoB cells to TLR stimulation in vitro. Accordingly, after short term stimulation with LPS, MZB cells show an enhanced proliferation and PC-differentiation in comparison to FoB cells (Martin et al., Immunity, 2001; Oliver et al., JI, 1999; Fairfax et al., Semin Immunol., 2008; Genestier et al., JI, 2007, Meyer-Bahlburg et al., JI, 2009).*
- *In contrast to FoB cells, MZB cells rapidly produce plasmablasts upon immunization with TI-antigens within 3 days (Martin et al., Immunity, 2001).*

3. Reg. the adoptive transfer experiments with FoB cells in the immunocompetent mice (Fig. 6): why exactly can it be excluded that the MZB do not develop from a very minor contaminating MZB population? Especially since there seems to be some proliferation involved (see criticism #1 above).

The following points support the assumption that the newly generated MZB cells are not derived from contaminating MZB cells

- In most of our experiments, we did not detect any MZB cells at day 1 after transplantation.
- MZB cells developed gradually after transplantation via a MZP phenotype. Early after transplantation, mainly MZB progenitor cells are visible, while mature MZB cells appeared later (Figures 6 and 7). If proliferation of transplanted mature MZB cells occurs, we would expect the expansion of MZB cells without this intermediate stage.
- New data (see our answer to point 2 of reviewer 3) revealed that within a time window of 7 days only around 10% of MZB cells incorporate BrdU, suggesting that wt MZB cells do not show a very strong proliferation.
- To further support our arguments, we made a calculation based on the MZB cell percentages of Fig. 6c and Supplementary Fig. 6c (see also source data in the accompanying Excel-file). In comparison to day 1 (mean: 0.063% MZBs within CD45.1⁺ cells) MZB cells were 69-fold enriched at day 4 (mean: 3.7% MZBs within CD45.1⁺ cells). Assuming that all CD45.1 positive MZB cells are derived from contaminating MZBs, CD23^{low}CD21^{high} cells must have divided around 6 times without any cellular loss to achieve this enrichment. These are very rapid generation times (< 12 hours). However, our BrdU incorporation data argue only for a modest proliferation of a subpopulation of the MZBs. Thus, it is very unlikely that the few contaminating MZBs proliferate so vigorously that they are the main source for the newly generated MZB cells in the transplantation experiment.
- Nevertheless, we agreed with the reviewer's comment and we believe that it is an important point to exclude the possibility that the newly generated MZB cells after transplantation of FoB cells could result from a small proportion of contaminating MZB cells. For this reason, we decided to compare the engraftment efficiency of MZB and FoB cells to rule out a massive proliferation of MZB cells after adoptive transfer. MZB cells were purified in parallel to FoB cells by cell sorting from the same pool of donor splenocytes. Naturally, MZB percentages among splenocytes and the subsequent cell yield after sorting was much lower compared to FoB cells. Therefore, the number of transferred cells was adapted for the MZB cell transplantation experiments. 5x10⁶ purified FoB and 1x10⁶ MZB cells were transferred into recipient sibling mice and analyzed 14 days after transfer. Staining for CD45.1⁺ engrafted cells showed that while 0.11% CD45.1⁺ cells were found among B cells after FoB transplantation, 0.022% and 0.023% were found after MZB transfer, thus exactly matching the relative differences in initially transferred cell numbers. These results prove that naïve mature MZB cells do not exhibit differences in engraftment efficiency compared to their FoB counterparts or showed spontaneous proliferation after transfer.

Engraftment efficiencies of purified FoB and MZB cells FoB and MZB cells were purified from the same pool of splenocytes. 5×10^6 FoB cells and 1×10^6 MZB cells were transferred and the engraftment was analysed 14 days after transfer.

We believe that these additional data convincingly show that the newly generated MZB cells after transfer of FoB cells are not derived from a massive expansion of contaminating MZB cells. We show these data only as additional information for reviewer #3 without including them in the manuscript. However, if the reviewer thinks this should be included, we would add it as supplemental information.

4. On page 5, I suggest to exchange "...continuously approached..." (this expression seems somewhat convoluted) by "...gradually acquired..." a MZB cell surface phenotype.

We have changed the text according to the suggestion of the reviewer.

5. On page 6, one should probably add after "...induces a phenotypic conversion from FoB towards MZB cell..." "..., at least with regard to the markers analyzed, ...". The RNA-seq analysis which justifies this sentence comes only later.

We have changed the text according to the suggestion of the reviewer.

REVIEWERS' COMMENTS

Reviewer #2 (Remarks to the Author):

The authors have adequately addressed my earlier concerns.

Reviewer #3 (Remarks to the Author):

In their revision, the authors have more than satisfactorily addressed and answered the issues I had regarding clarity or interpretation of data. Additional experiments have been performed.

The experiments performed to address my point #3 do not need to be added to the supplementary information as other strong arguments have been provided that rule out my initial concern. But it is good to know once and for all that outgrow of a (potentially contaminating) minor marginal zone B-cell population is not an issue.

The revisions have further strengthened the manuscript, which I now consider acceptable for publication.